# Antitrypanosomal Activity of Sesquiterpene Lactones from *Helianthus tuberosus* L. Including a New Furanoheliangolide with an Unusual Structure

**DOI:** 10.3390/molecules24061068

**Published:** 2019-03-18

**Authors:** Anna Galkina, Nico Krause, Mairin Lenz, Constantin G. Daniliuc, Marcel Kaiser, Thomas J. Schmidt

**Affiliations:** 1Institute of Pharmaceutical Biology and Phytochemistry (IPBP), University of Münster, PharmaCampus–Corrensstrasse 48, D-48149 Münster, Germany; no4k-a@yandex.ru (A.G.); Nico1Krause@gmx.de (N.K.); mailen@uni-muenster.de (M.L.); 2Institute of Organic Chemistry, University of Münster, Corrensstraße 40, D-48149 Münster, Germany; constantin.daniliuc@uni-muenster.de; 3Swiss Tropical and Public Health Institute (Swiss TPH), Socinstrasse 57, CH-4051 Basel, Switzerland; marcel.kaiser@unibas.ch; 4University of Basel, Petersplatz 1, CH-4003 Basel, Switzerland

**Keywords:** sesquiterpene lactone, furanoheliangolide, X-ray crystallography, antiprotozoal activity, *Trypanosoma brucei rhodesiense*, *Trypanosoma cruzi*, *Leishmania donovani*, *Plasmodium falciparum*, cytotoxicity

## Abstract

As part of our efforts to exploit the antitrypanosomal potential of sesquiterpene lactones (STL) from *Helianthus tuberosus* L. (Asteraceae), besides the known 4,15-*iso*-atriplicolide tiglate, -methacrylate and -isobutyrate, a hitherto unknown STL was isolated. Its structure was solved by extensive NMR measurements and confirmed by single crystal X-ray crystallography. This novel compound is a structural analog 4,15-*iso*-atriplicolide tiglate that possesses the same basic furanoheliangolide skeleton but differs in the position of the oxo function which is at C-2 instead of C-1, as well as in the fact that the oxygen atom of the furanoid ring is part of a hemiketal structure at C-3 and a double bond between C-5 and C-6. For this new STL we propose the name heliantuberolide-8-*O*-tiglate. Its activity against *Trypanosoma brucei rhodesiense* (causative agent of East African Human Typanosomiasis, *Trypanosoma cruzi* (Chagas Disease), *Leishmania donovani* (Visceral Leishmaniasis) and *Plasmodium falciparum* (Tropical Malaria) as well as cytotoxicity against rat skeletal myoblasts (L6 cell line) was determined along with those of the hitherto untested 4,15-*iso*-atriplicolide methacrylate and isobutyrate. In comparison with the *iso*-atriplicolide esters, the new compound showed a much lower level of bioactivity.

## 1. Introduction

In a previous communication, we reported on the identification of furanoheliangolide-type sesquiterpene lactones (STLs) as strong trypanocides with high activity and selectivity against *Trypanosoma brucei rhodesiense*, the causative agent of East African Human Typanosomiasis [1]. A compound from the aerial parts of *Helianthus tuberosus* L. (Jerusalem Artichoke; Asteraceae), 4,15-*iso*-atriplicolide tiglate was found particularly active with an IC_50_ value as low as 15 nM against the parasite. In an attempt to isolate a higher quantity of this STL, as well as further related compounds from this plant, we found, besides the mentioned tiglate, the methacrylate and isobutyrate analogues along with a further, hitherto undescribed STL. Here we report on the structure elucidation, as well as the results of antiprotozoal tests of this new STL and the hitherto untested 4,15-*iso*-atriplicolide esters.

## 2. Results and Discussion

### 2.1. Isolation of STLs from *H. tuberosus* and Structure Elucidation of a New Furanoheliangolide

From the dichloromethane extract obtained from aerial parts of *Helianthus tuberosus* L. we obtained, by column chromatography on silica followed by preparative HPLC, the known 4,15-*iso*-atriplicolide tiglate (**1**) as well as the analogous esters with methacrylic (**2**) and isobutryric acid (**3**) [2]. The structures are depicted in Figure 1. Additionally, a fourth compound (**4**) was isolated, which had not been described before. Its molecular mass as determined by QqTOF MS was higher than that of **1** by 18 units, leading to an elemental formula of C_20_H_24_O_7_, indicating the addition of H_2_O to **1**. This was in agreement with the considerably shorter HPLC retention time of **4** on a C18 column in comparison with the *iso*-atriplicolide esters. The ^13^C-NMR spectrum (Table 1) of **4** confirmed the number of carbon atoms and, in combination with the ^1^H/^13^C HSQC experiment, showed the presence of four methyl, three methylene (1× sp^2^, 2× sp^3^), five methine (2× sp^2^, 3× sp^3^) and eight quaternary carbons. Of the latter, the characteristic chemical shift ranges indicated the presence of one ketone, two ester or lactone, three olefinic as well as two sp^3^ carbons substituted with oxygen of which one resonated at δ 99.5 indicating a (hemi-) acetal or ketal structure. A large number of signals, including those of the tigloyl ester moiety were very similar to those of **1** but major differences were found with respect to the structural fragment comprising positions C-1 to C-5. The connectivity in this part of the molecule was elucidated by analysis of the ^1^H/^13^C-HMBC spectrum. Instead of the α, β, γ, δ-unsaturated ketone structure constituted by C-1, 2, 3, 4, and 15 in compound **1**, the keto carbon of compound **4** (δ 210.2) was not conjugated with a double bond but in the neighborhood of an sp^3^ methylene group (δ_C_ 45.4, δ_H_ 3.14 and 2.68), which also showed correlations with the above mentioned dioxygenated quaternary acetal or ketal carbon at δ 99.5 (C-3), as well as with the other oxygen substituted quaternary carbon at δ 78.7 (C-10), indicating an ether function at this position. The former was found to be connected to a methyl substituted C=C double bond as evident from its HMBC correlation with an olefinic methine proton at δ 5.70 and a methyl group at δ 1.94 in the ^1^H-NMR spectrum (^1^H-NMR data see Table 1). This indicated that as a further difference to **1**, C-4 and C-5 are connected by a double bond whereas C-15 is the mentioned methyl group. Interestingly, signals of the protons at C-9 (δ 2.37, 2.29), as well as the corresponding carbon signal (δ 43.4), showed clear HMBC correlations with the signals of the methylene group (δ 45.4) next to the keto carbon. This methylene was hence found to represent C-1 and not C-2 as in compound **1** and vice versa for the keto group. Overall, the depicted connectivity of the whole molecule could be unambiguously established by the HMBC correlations. Furthermore, the relative stereochemistry at most stereocenters was clearly the same as in **1,** as became evident by the very similar coupling constants in the proton signals over a large part of the molecule and by means of a ^1^H/^1^H-NOESY spectrum. Nevertheless, some ambiguity remained with respect to the configuration at the hemiketal center, C-3, since the OH proton (δ 2.72; no exchange for deuterium and thus obviously engaged in a rather stable hydrogen bond) only showed an NOE interaction with the methyl group 15, which might be possible in both, the α as well as the β-position. Fully optimized 3D molecular models of both stereoforms indicated, however, that this OH group should show NOEs with H-6 and H-9 β in case of the β-oriented OH group, which are not possible in case of the α-hydroxy epimer and were not observed in the spectrum. Thus, the configuration at C-3 was very likely represented by the α-OH epimer as depicted. This result, as well as all other structural assignments, could be confirmed by single crystal X-ray crystallographic analysis. Compound **4** crystallized in an orthorhombic crystal system with four molecules in the chiral space group *P*2_1_2_1_2_1_. The absolute configuration can be determined as (3*S*,6*R*,7*S*,8*R*,10*R*) with the Flack parameter refined to 0.09(6) and Hooft parameter refined to 0.06(7). The crystal structure thus obtained is shown in Figure 2.

It is noteworthy that an isomeric 1-oxo-2-methylene-isomer of this compound has previously been described as a constituent of *H. tuberosus* [2]. However, at that time, the assignment was based solely on ^1^H-NMR data, and obviously by means of analogy to the *iso*-atriplicolide and budlein-type compounds from the same plant, the oxo-function was assigned at C-1. The ^1^H-NMR data alone, however, would not allow a distinction of the position of the keto group. The reported ^1^H-NMR data for this STL (compound 17 in ref. [2]) are almost identical to those of compound **4** so that it is likely that the same compound was isolated but the structure not fully correct. In fact, no previous reports exist on a furanoheliangolide of this type with an oxo-function at C-2 instead of C-1. For this new STL, the generic name *heliantuberolide*-8-*O*-tiglate is proposed.

### 2.2. Antiprotozoal and Cytotoxic Activity of the Isolated STLs

The activity of 4,15-*iso*-atriplicolide tiglate (**1**) against *T. brucei rhodesiense* (*Tbr*), *T. cruzi* (*Tc*), *Plasmodium falciparum* (*Pf*) and its cytotoxicity against L6 rat skeletal myoblasts was previously reported [1] and are repeated in Table 2 for easier comparison with the compounds isolated for the present study.

Sesquiterpene lactones have in many cases shown prominent antiprotozoal activity, in particular against *Tbr* [1,3,4,5,6,7]. This activity has been related to the presence of potentially reactive enone systems and our structure-activity relationship (SAR) studies have demonstrated that the presence of more than one such structure element in many cases confers a much higher level of activity [1,7,8]. In case of the exceptionally strong trypanocide 1, this structural requirement is represented by the α-methylene-γ-lactone and the dienone system. The fact that the corresponding esters, 2 and 3, which share these structural features and are identical with 1 except for the ester moieties, are still quite active but at a lower level shows that the ester moiety in this class of compounds has a modulating influence on the bioactivity. On the other hand, the new STL **4** is rather similar in size and shape to 1 (including the same ester moiety) but about 60 times less active which nicely confirms the dependence of strong activity on the presence of the conjugated dienone in addition to the methylene lactone group. The present finding thus confirms our earlier observations on SAR of STLs. Since the tigloyl group of 1 apparently confers higher activity than the smaller C4-acyl groups of 2 and 3, it appears interesting to test analogues with even larger ester moieties in the future. Investigations on the mechanism of action of compounds **1**–**3** are in progress.

## 3. Materials and Methods

### 3.1. General Methods

#### 3.1.1. Ultra-High Performance Liquid Chromatography/Mass Spectrometry (UHPLC/MS)

The dichloromethane (DCM) extract, collected fractions and isolated compounds were dissolved in methanol at concentrations of 10 mg/mL and 0.1 mg/mL for crude extract or fractions, and for pure compounds, respectively. Chromatographic separations were performed on a Dionex Ultimate 3000 RS Liquid Chromatography System (Idstein, Germany) with a Dionex Acclaim RSLC 120, C18 column (2.1 × 100 mm, 2.2 µm) using a binary gradient (A: water with 0.1% formic acid; B: acetonitrile with 0.1% formic acid) at 0.8 mL/min: 0–9.5 min: linear from 5% B–100% B; 9.5–12.5 min: isocratic 100% B; 12.5–12.6 min: linear from 100% B down to 5% B; 12.6–15 min: isocratic 5% B. The injection volume was 5 µL. Eluted compounds were detected using a Dionex Ultimate DAD-3000 RS over a wavelength range of 200–400 nm and a Bruker Daltonics micrOTOF-QII quadrupole/time-of-flight mass spectrometer (Bremen, Germany) equipped with an Apollo electrospray ionization source in positive mode at 5 Hz over a mass range of *m*/*z* 50–1000 using the following instrument settings: nebulizer gas nitrogen, 5 bar; dry gas nitrogen, 9 L/min, 220 °C; capillary voltage 4500 V; end plate offset −500 V; transfer time 70 µs; collision gas nitrogen; collision energy and collision radio frequency (RF) settings were combined for each single spectrum of 1000 summations as follows: 250 summations with 20% base collision energy; 130 Vpp+ 250 summations with 100% base collision energy; 500 Vpp+ 250 summations with 20% base collision energy; and 130 Vpp + 250 summations with 100% base collision energy and 500 Vpp. Base collision energy was 50 eV for precursor ions with a m/z less than 500 and then linearly interpolated against m/z up to a maximum of 70 eV for precursor ions with a *m*/*z* of up to 1000. Internal dataset calibration (HPCmode) was performed for each analysis using the mass spectrum of a 10 mM solution of sodium formate in 50% isopropanol that was infused during LC re-equilibration using a divert valve equipped with a 20-µL sample loop. The retention times and mass spectral data of compounds **1**–**4** are reported below (analytical data).

#### 3.1.2. Preparative HPLC

Preparative HPLC separations were performed on a Jasco (Groß-Umstadt, Germany) preparative HPLC system (pump: PU-2087 plus; diode array detector MD 2018 plus; column thermostat CO 2060 plus; autosampler AS 2055 plus; LC Net II ADC Chromatography Data Solutions; sample injection loop: 2000 µL) on a preparative reverse phase column Reprosil 100 C-18 (5 µm, 250 mm × 20 mm, Macherey-Nagel, Düren, Germany) with binary gradients of the mobile phase. The optimized mobile phase was composed of water (A) and methanol (B) using the following gradient conditions: 40–70% of B (0 to 35 min), 70–100% of B (35 to 38 min), 100% of B (38 to 43 min) and another three minutes to return to the initial conditions and five for re-equilibration. A flow rate of 10 mL/min and a column temperature of 30 °C were used in all separations. Chromatograms were recorded at 215, 260 and 280 nm. The retention times of compounds **1**–**4** in this system are reported below (3.3 Extraction and isolation).

#### 3.1.3. NMR Spectroscopy

NMR spectra (^1^H, ^13^C, ^1^H/^1^H-COSY, ^1^H/^1^H-NOESY, ^1^H/^13^C-HSQC, and ^1^H/^13^C-HMBC) were recorded on a 600 MHz Agilent DD2 spectrometer. The spectra were obtained at 298 K in CDCl_3_. The CDCl_3_ solvent signals (^1^H: 7.260 ppm and ^13^C: 77.000 ppm) were used to reference the spectra. MestReNOVA v. 11 (Mestrelab Research, Chemistry Software Solutions, Santiago de Compostela, Spain) software was used to process and evaluate the spectra.

#### 3.1.4. X-ray Diffractometry

X-ray data for compound **4** were collected at 115K on a Bruker D8-Venture dual diffractometer with Cu Micro source and PHOTON 100 detector (Bruker, Karlsruhe, Germany). Data collection was done using *APEX*3 software (V2016.1-0) [9]. Data were corrected for absorption effects using the multi-scan method applied by the SADABS program (V2014/7) [9]. The structure was solved by direct method with *SHELXT*-2015 [10] and refined by a full-matrix least-squares treatment of *F^2^* using the *SHELXL*-2015 [10] refinement package. Graphics were created with *XP*-2015 [9]. *R*-values are given for observed reflections, and *wR*^2^ values are given for all reflections.

CCDC-1898576 for compound 4 contains the Appendix A for this paper. These data can be obtained free of charge from the Cambridge Crystallographic Data Centre via www.ccdc.cam.ac.uk/data_request/cif (21 February 2019).

### 3.2. Plant Material

*Helianthus tuberosus* L. was cultivated in the garden of the Institute of Pharmaceutical Biology and Phytochemistry (IPBP), University of Münster. The plant material (aerial parts, pre-flowering stage) for the current study was harvested in October 2017 and dried at 40 °C in a drying oven. A voucher specimen is deposited in the IPBP herbarium with the voucher code 475.

### 3.3. Extraction and Isolation

1 kg of dried and powdered plant material was exhaustively extracted in 8 portions (≈125 g each) in a soxhlet apparatus with DCM. After rotary evaporation under reduced pressure, 37 g of crude extract (3.7%) were obtained. Four portions (3 g each) of the extract were separated under near identical conditions by ambient pressure CC on silica gel using a glass column (80 × 5 cm) and 300 g of the stationary phase in each case. Mixtures of n-hexane (hex) and ethyl acetate (EtOAc) of increasing polarity were used as mobile phase and similar eluates combined in 13 fractions (F1-F13) after TLC control (silica 60 F 254, hex:EtOAc mixtures). The fractions were eluted with hex:EtOAc 90:10 (1 L); 80:20 (1L; F1); 70:30 (2 L; F2- F3); 60:40 (2.5 L; F4-F6); 50:50 (2 L; F7-F10); 40:60 (1 L; F11); 0:100 (1 L; F12-F13). Representative fractions were then analyzed by LC/MS for the occurrence of the target compounds (4,15-*iso*-atriplicolide esters **1**–**3**) which were detected in F4–F6. Their isolation was achieved by preparative HPLC (see general methods) from the mentioned fractions which contained all compounds in variable amounts. Thus, in total, 49.1 mg of **1** (tR = 32.9 min), 24.4 mg of **2** (tR = 28.6 min), 13.4 mg of **3** (tR = 30.3 min) and 31.7 mg of compound **4** (tR = 31.7 min) were obtained as colorless white solids. For the crystallographic analysis, compound **4** was crystallized from hex:EtOAc 20:80 by cooling a concentrated solution in a refrigerator.

### 3.4. X-ray Crystallographic Analysis of Compound ***4***

A colorless prism-like specimen of C_20_H_24_O_7_, approximate dimensions 0.089 mm × 0.129 mm × 0.245 mm, was used for the X-ray crystallographic analysis. The X-ray intensity data was measured.

A total of 1356 frames were collected. The total exposure time was 17.59 h. The frames were integrated with the Bruker SAINT software package using a wide-frame algorithm as part of the APEX3 software suite (Bruker, Madison, WI, USA). The integration of the data using an orthorhombic unit cell yielded a total of 18431 reflections to a maximum Θ angle of 68.37° (0.83 Å resolution), of which 3487 were independent (average redundancy 5.286, completeness = 99.7%, R_int_ = 4.05%, R_sig_ = 3.07%) and 3336 (95.67%) were greater than 2 σ(F^2^). The final cell constants of a = 7.4900(2) Å, b = 15.4129(3) Å, c = 16.6182(4) Å, volume = 1918.45(8) Å^3^, are based upon the refinement of the XYZ-centroids of 9965 reflections above 20 σ (I) with 7.823° < 2 Θ < 136.6°. Data were corrected for absorption effects using the multi-scan method with the SADABS program (Bruker, Madison, WI, USA). The ratio of minimum to maximum apparent transmission was 0.918. The calculated minimum and maximum transmission coefficients (based on crystal size) are 0.8240 and 0.9300.

The structure of compound **4** was solved and refined using the Bruker SHELXTL software package [10], using the chiral space group *P*2_1_2_1_2_1_, with *Z* = 4 for the formula unit C_20_H_24_O_7_. The final anisotropic full-matrix least-squares refinement on F^2^ with 249 variables converged at R1 = 2.91%, for the observed data and wR2 = 7.19% for all data. The goodness-of-fit was 1.036. The largest peak in the final difference electron density synthesis was 0.137 e^−^/Å^3^ and the largest hole was −0.166 e^−^/Å^3^ with an RMS deviation of 0.040 e^−^/Å^3^. On the basis of the final model, the calculated density was 1.303 g/cm^3^ and F(000), 800 e^−^. The Flack parameter was refined to 0.09(6).

All details and results are reported in Appendix A.

### 3.5. Analytical Data

(3a*S*,4*R*,6*R*,11a*R*)-6-methyl-3,10-dimethylene-2,7-dioxo-2,3,3a,4,5,6,7,10,11,11a-decahydro-6,9-epoxycyclodeca[b]furan-4-yl (*E*)-2-methylbut-2-enoate (4,15-*iso*-atriplicolide tiglate, **1**) UHPLC/+ESI-QTOF MS: tR 7.66 min, MS (*m*/*z*): 359.1492 [M + H]^+^; 376.1751 [M + NH_4_]^+^; 381.1301 [M + Na]^+^; calcd. for C_20_H_23_O_6_^+^: 359.1492, C_20_H_26_NO_6_^+^: 376.1755, C_20_H_22_O_6_Na^+^: 381.1309. ^1^H and ^13^C-NMR identical with literature [11].

(3a*S*,4*R*,6*R*,11a*R*)-6-methyl-3,10-dimethylene-2,7-dioxo-2,3,3a,4,5,6,7,10,11,11a-decahydro-6,9-epoxycyclodeca[b]furan-4-yl methacrylate (4,15-*iso*-atriplicolide methacrylate, **2**) UHPLC/+ESI-QTOF MS: tR 7.30 min, MS (*m*/*z*): 345.1325 [M + H]^+^; 362.1573 [M + NH_4_]^+^; 367.1151 [M + Na]^+^; calcd. for C_19_H_21_O_6_^+^: 345.1333, C_19_H_24_NO_6_^+^: 362.1604, C_19_H_20_O_6_Na^+^: 367.1152. ^1^H-NMR identical with literature [2].

(3a*S*,4*R*,6*R*,11a*R*)-6-methyl-3,10-dimethylene-2,7-dioxo-2,3,3a,4,5,6,7,10,11,11a-decahydro-6,9-epoxycyclodeca[b]furan-4-yl methacrylate (4,15-*iso*-atriplicolide isobutryrate, **3**) UHPLC/+ESI-QTOF MS: tR 7.43 min, MS (*m*/*z*): 347.1481 [M + H]^+^; 364.1745 [M + NH_4_]^+^; 369.1306 [M + Na]^+^; calcd. for C_19_H_23_O_6_^+^: 347.1489, C_19_H_26_NO_6_^+^: 364.1760, C_19_H_22_O_6_Na^+^: 369.1309. ^1^H identical with literature [2].

(3a*R*,4*R*,6*R*,9*S*,11a*R*,*Z*)-9-hydroxy-6,10-dimethyl-3-methylene-2,8-dioxo-2,3,3a,4,5,6,7,8,9,11a-decahydro-6,9-epoxycyclodeca[b]furan-4-yl (*E*)-2-methylbut-2-enoate (Heliantuberolide-8-*O*-tiglate, **4**) UHPLC/+ESI-QTOF MS: tR 7.44 min, MS (*m*/*z*): 377.1579 [M + H]^+^; 394.1854 [M + NH_4_]^+^; 399.1404 [M + Na]^+^; calcd. for C_20_H_25_O_7_^+^: 377.1595, C_20_H_28_NO_7_: 394.1866^+^, C_20_H_24_O_7_Na^+^: 399.1414. ^1^H and ^13^C-NMR see Table 1.

### 3.6. Biological Assays

Tests for antiprotozoal activities and cytotoxicity were carried out using established standard protocols at the Swiss Tropical and Public Health Institute (Swiss TPH, Basel, Switzerland). The bioassays and determination of the IC_50_ values were performed as described previously [8], except in the case of *P. falciparum*, where the NF54 strain instead of the K1 strain was used.

The compounds used as positive controls in the various bioassays (see Table 1) were of commercial origin, with the exception of melarsoprol, which was a gift from the WHO. Their purity (generally >95%) was specified by the manufacturers.

The purity of test compounds was assessed by UHPLC/MS and ^1^H-NMR analyses and found to be >95% in all cases.

## 4. Conclusions

The present study has shown that the ester moiety has a modulatory influence on the antitrypanosomal potency of 4,15-*iso*-atriplicolide derivatives and gave first insights into the structure-activity relationship indicating that a larger and more lipophilic ester group enhances this activity. Thus, the tiglate **1** is still the most potent STL found so far against *Tbr*. On the other hand, the relatively low activity of the new heliantuberolide-8-*O*-tiglate **4**, which lacks the dienone structure and contains a hitherto undescribed hemiketal structure with a keto function at C-2 instead of C-1, clearly shows the essential contribution of the α, β, γ, δ-unsaturated ketone to the antitrypanosomal effect.

## Figures and Tables

**Figure 1 molecules-24-01068-f001:**
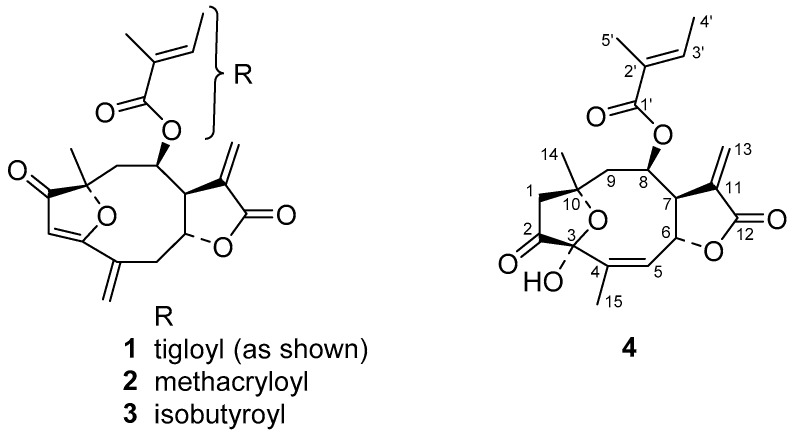
Structures of the isolated STLs (sesquiterpene lactones).

**Figure 2 molecules-24-01068-f002:**
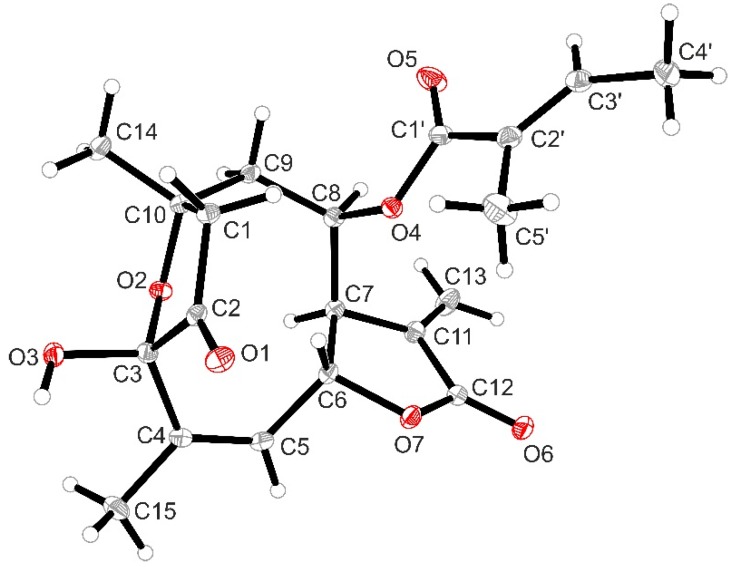
*XP* diagram representing the crystal structure of the new STL **4**. Thermal ellipsoids are shown at 30% probability.

**Table 1 molecules-24-01068-t001:** ^13^C and ^1^H NMR data of the new STL (sesquiterpene lactones) **4** (150 and 600 MHz, respectively, CDCl_3_); all assignments were confirmed by 2D homo- and heteronuclear correlation spectra.

Position	δ_C_	δ_H_	Mult.	J (Hz)
1	45.4	3.142	d	19.9
		2.684	d	19.9
2	210.2	--		
3	99.5	--		
4	138.9	--		
5	131.2	5.704	dq	6.5; 1.5
6	76.6	5.131	ddq	6.5; 1.5; 1.5
7	48.7	3.612	m	
8	77.2	5.275	ddd	3; 4; 1
9	43.4	2.370	dd	3.3; 16.2
		2.292	dd	4.3; 16.2
10	78.7	--		
11	138.2	--		
12	169.4	--		
13	124.8	6.329	d	2.5
		5.742	d	2.2
14	32.5	1.506	s	
15	20.1	1.939	dd[t]	1.5; 1.5
1′	166.6	--		
2′	127.9	--		
3′	139.6	6.758	qq	1.4; 7.0
4′	14.8	1.787	dq	1.1; 7.0
5′	12.23	1.766	dq	≈1.2
3-OH	--	2.724	br s	

**Table 2 molecules-24-01068-t002:** Antiprotozoal and cytotoxic activity data of the isolated STLs (IC_50_ values in µM; means of two independent determinations ± deviation from the mean).

Compound	*Tbr*	*Tc*	*Ld*	*Pf*	L6
**1 ^a^**	0.015 ± 0.003	3.7 ± 1.3	n.t.	1.0 ± 0.2	1.2 ± 0.5
**2**	0.077 ± 0.002	1.6 ^b^	0.83 ± 0.47	1.4 ± 0.1	0.52 ± 0.13
**3**	0.26 ± 0.10	3.1 ^b^	0.37 ± 0.19	0.83 ± 0.04	0.88 ± 0.26
**4**	0.92 ± 0.26	5.7 ± 0.6	2.0 ± 0.3	5.5 ± 0.3	3.9 ± 1.2
**Pos. Contr.**	0.008 ± 0.003 ^c^	1.45 ± 0.18	1.09 ± 0.21 ^e^	0.0012 ± 0.003 ^f^	0.024 ± 0.002 ^g^

^a^ data from [1] are repeated here for easier comparison; ^b^ results of single determinations; ^c^ melarsoprol; ^d^ benznidazole; ^e^ miltefosine; ^f^ chloroquine; ^g^ podophyllotoxin; n.t. not tested.

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
