# Peer review of "Antitrypanosomal Activity of Sesquiterpene Lactones from Helianthus tuberosus L. Including a New Furanoheliangolide with an Unusual Structure"

_molecules, 2019, doi:10.3390/molecules24061068_

Round 1

Reviewer 1 Report

Dear Authors,

This is a well written manuscript  by Anna Galkina and collaborators that presents the in vitro antiparasitic activity of four STLs against protozoan parasites responsible for important NTDs. Compound 1 showed high specific activity against T. brucei and based on the chemical structure of the molecules it was possible to demonstrate  the essential contribution of the α, β, γ and δ unsaturated ketone to the anti T. brucei activity.

For review:

 In the Table 2 the SD value of Benznidazole was forgotten?

Is there no SD for compound 2 and 3 against T. cruzi?

as well as, their cytotoxicity on L6 myoblast cells. 

Author Response

Reviewer 1

This is a well written manuscript  by Anna Galkina and collaborators that presents the in vitro antiparasitic activity of four STLs against protozoan parasites responsible for important NTDs. Compound 1 showed high specific activity against T. brucei and based on the chemical structure of the molecules it was possible to demonstrate  the essential contribution of the α, β, γ and δ unsaturated ketone to the anti T. brucei activity.

For review:

 In the Table 2 the SD value of Benznidazole was forgotten?

We thank the reviwer for being so scrutinous. This was a typographic error but we noticed then that the data reported for the positive controls were from a previous measuring series and reported in µg/mL, i.e. had not been transformed to µM. This has now been corrected.

Is there no SD for compound 2 and 3 against T. cruzi?

These two measurements were only performed once so no deviation can be reported. This fact is clearly annotated with a footnote in the table.

as well as, their cytotoxicity on L6 myoblast cells.

It is not clear what the reviewer means here. The sentence is incomplete.

We thank the reviewer for her/his time and efforts!

Reviewer 2 Report

This paper reported  sesquiterpene lactones from Helianthus tuberosus L. (Asteraceae) with antitrypanosomal activity. Overall, it is a well-written paper, but some minor changes are required.

(1) Line 53, change "QqTOF MS" to "Q-TOF MS".

(2) Line 55, change "RP-18 column" to "RP-C18 column".

(3) Normally,  the NOE correlations of the exchangeable proton such as OH was not reliable for configuration determination. May consider delete the discussion of OH group NOEs of compound 4, and directly determine the configuration by  single crystal X-ray crystallographic analysis.

(4) Lines 144 and 145, please correct "C18 column (2.1 _ 100 mm, 2.2 _m)".

(5) Lines 152 and 161,  correct  "9 L/min, 220 _C". and "a 20-_L sample loop".

(6) Line 248, correct  "C20H26NO6" and "C20H22O6Na" to "C20H26NO6+" and "C20H22O6Na+". Same as the errors in lines 253, 258 and 263.

Author Response

Reviewer 2

This paper reported  sesquiterpene lactones from Helianthus tuberosus L. (Asteraceae) with antitrypanosomal activity. Overall, it is a well-written paper, but some minor changes are required.

(1) Line 53, change "QqTOF MS" to "Q-TOF MS".

The correct nomenclature is QqTOF MS. The reviewer might consult, e.g. Hopfgartner G, Mass Spectrometry in Bioanalysis – Methods, Prinicples and Applications. In Wanner K, Höfner G (Eds.) Mass Spectrometry in Medicinal Chemistry. Wiley VCH, 2007. In short, the Qq terminology indicates that the MS system hast two quadrupoles, one for actively selecting ion species (Q) and the other (q) serving merely as a collision module. No change applied.

(2) Line 55, change "RP-18 column" to "RP-C18 column".

The correct nomenclature is either RP-18 or C18. The reviewer might e.g. google for RP-18 (sometimes RP18) where a lot of hits will appear with typical manufacturers‘ column names, all with RP18 and without the „C“. Since the Dionex column used by us is called Dionex Acclaim RSLC 120, C18 (see experimental section), we have replaced the RP-18 by C18 (which has, of course, the same meaning).

(3) Normally,  the NOE correlations of the exchangeable proton such as OH was not reliable for configuration determination. May consider delete the discussion of OH group NOEs of compound 4, and directly determine the configuration by  single crystal X-ray crystallographic analysis.

We disagree with the reviewer. It is a well known fact that O-H protons engaged in hydrogen bonds have a reduced tendency to be exchanged by deuterons. Therefore, this is even an important fact for readers with a background in NMR spectroscopy. Deleting this part would hence mean to omit an important piece of information. On the contrary, we have therefore added a short note on this observation.

(4) Lines 144 and 145, please correct "C18 column (2.1 _ 100 mm, 2.2 _m)".

The characters were missing: 2.1 x 100 mm, 2.2 µM. This was corrected.

(5) Lines 152 and 161,  correct  "9 L/min, 220 _C". and "a 20-_L sample loop".

This was corrected. 220°C; 20 µL

(6) Line 248, correct  "C20H26NO6" and "C20H22O6Na" to "C20H26NO6+" and "C20H22O6Na+". Same as the errors in lines 253, 258 and 263.

This was corrected in all instances.

We thank the reviewer for her/his time and efforts!